Host-Microbe Biology

# Cyclophosphamide Increases *Lactobacillus* in the Intestinal Microbiota in Chickens

Dany Mesa,[a] Breno C. B. Beirão,[b] Eduardo Balsanelli,[a] Luiz Sesti,[c] Luiz F. Caron,[b] Leonardo M. Cruz,[a] Emanuel M. Souza[a]

[a]Department of Biochemistry and Molecular Biology, Federal University of Paraná, Curitiba, Brazil
[b]Department of Pathology, Federal University of Paraná, Curitiba, Brazil
[c]Ceva Animal Health, Campinas, Brazil

**ABSTRACT** Recent data in humans indicate that immunosuppression is correlated with shifts in the intestinal microbiota. However, the relationship between immunosuppression and intestinal microbiota has not been studied in chickens. Thus, we investigated the correlations between immune cells and intestinal microbiota by massive parallel sequencing of the 16S rRNA bacterial gene in chickens immunosuppressed with cyclophosphamide. The results showed correlations between peripheral immune cells and intestinal microbiota. Surprisingly, an increase in the abundance of intestinal *Lactobacillus* in the immunosuppressed chickens was observed. These birds also had low intestinal IgA antibody levels among other alterations in the microbiota. These shifts indicate a role of the immunity system in controlling the microbiota of birds.

**IMPORTANCE** Poultry production is a very intensive industry. Due to the substantial number of animals being raised by any one producer, even small variations in productivity lead to important economical outcomes. The intestinal microbiota of birds is under intense scrutiny by the immune system. Therefore, it is a factor that can influence the states of health and disease of the host. The body of knowledge on the interactions between these systems is gradually bringing practical guidance for poultry production.

**KEYWORDS** broiler, immunosuppression, *Faecalibacterium*, cloacal bursa, chicken, IgA, cecal microbiota

**P**oultry production is one of the largest food industries worldwide. In farms, the birds frequently endure conditions that can lead to immunosuppression, such as high mycotoxin concentrations in feed and infectious agents that target immune cells, such as infectious bursal disease virus (IBDV) (1). As a consequence, immunosuppressed chickens perform poorly, therefore affecting economic returns (2).

In addition, recent data indicate that immunosuppression is also correlated with characteristic changes in the intestinal microbiota in humans, likely due to damages to gut-associated lymphoid cells and tissues (3). This is important because the intestinal microbiota not only reflects the immune changes but also shapes the states of health and disease; for some immune-mediated conditions, alterations of the intestinal microbiota seem to be crucial aspects of the disease (3, 4).

Thus, the intestinal microbiota seems to interact directly with the immune system of the host, contributing to maintaining the integrity of the epithelial barrier and stimulating local and systemic immune interactions (5). These biological interactions are beginning to be studied in poultry science. For example, Luo et al. (6) observed an increase in immune proteins and changes in the intestinal microbiota in chickens treated with a probiotic, while Oakley and Kogut (7) showed a correlation between cytokines and intestinal microbiota in chickens.

However, the relationship between immunosuppression and intestinal microbiota

Address correspondence to Dany Mesa, dmesaf7@gmail.com, or Emanuel M. Souza, souzaem@ufpr.br.

Immunity system help to controlling the microbiota in birds

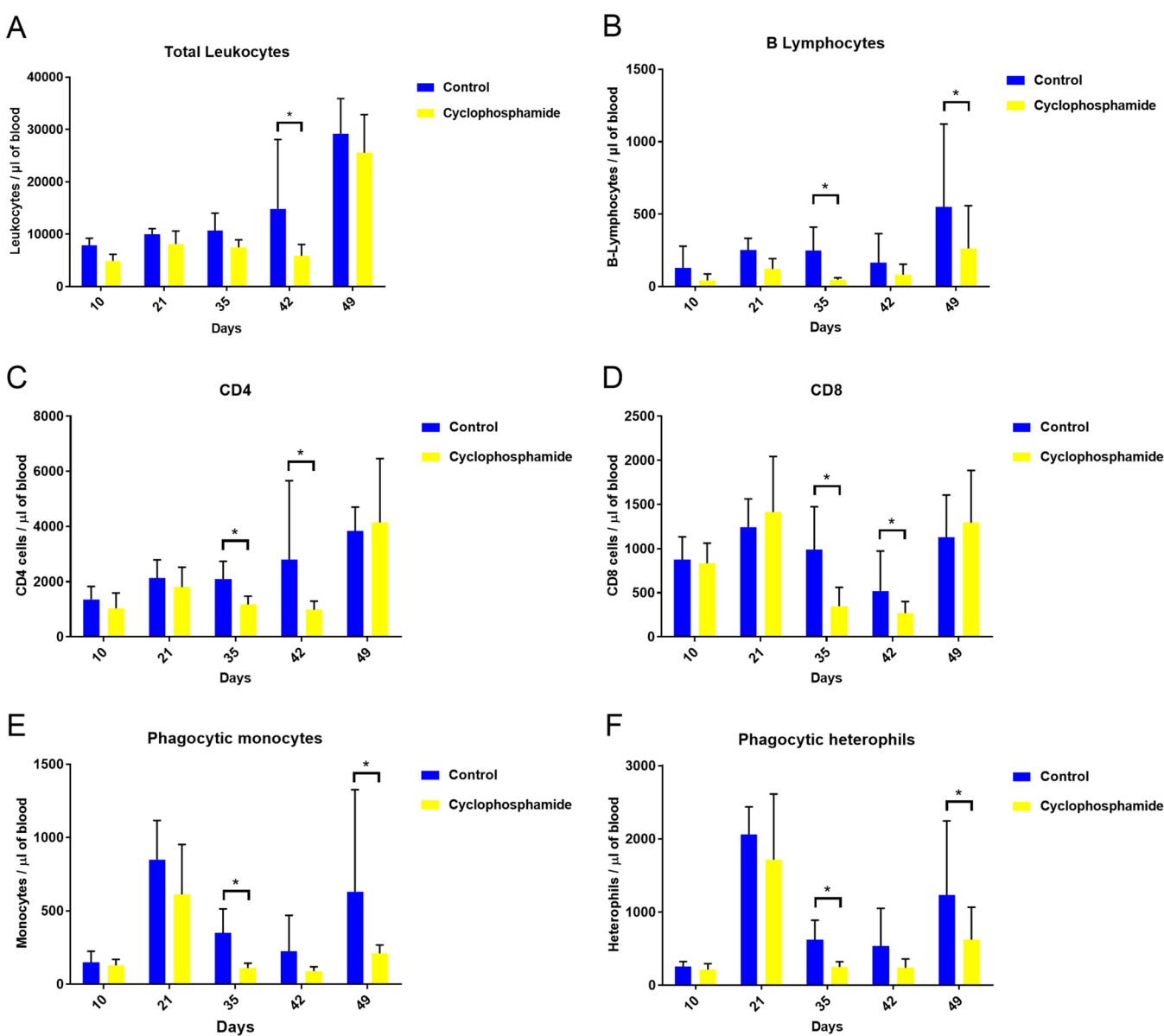

**FIG 1** Peripheral blood leukocytes were reduced by cyclophosphamide treatment. (A) Total leukocyte numbers. (B) B lymphocyte counts. Counts of CD45+ CD4+ (C) and CD45+ CD8+ (D) T lymphocytes, phagocytic monocytes (E), and heterophils (F) were determined by flow cytometry. Data are representative of 8 samples/group. Values represent the numbers of cells per microliter of blood (average ± standard deviation [SD]). *, $P < 0.05$ by Kruskal Wallis test.

has not been studied in chickens. To better understand this interaction, we investigated the correlations between immune cells and intestinal microbiota by massive parallel sequencing of the 16S rRNA gene in immunosuppressed chickens.

## RESULTS

Administration of cyclophosphamide caused a marked decrease in the number of peripheral blood leukocytes. Total leukocyte counts were decreased by approximately 65% at 42 days ($P < 0.05$) (Fig. 1A). B lymphocyte (Bu-1+) counts were significantly decreased by approximately 80% at 35 days and 65% at 49 days of age (Fig. 1B). Counts of other leukocytes in blood (CD45+ CD4+ and CD45+ CD8+, phagocytic monocytes, phagocytic heterophils, and CD45+ CD4+ T-cell receptor (TCR) Vβ1+ cells from 35 days of age) were also reduced in the group treated with cyclophosphamide (Fig. 1C and see Fig. S3 in the supplemental material). In addition, the cloacal bursa, which is the primary site for B lymphocyte development in birds, was significantly reduced in the chickens

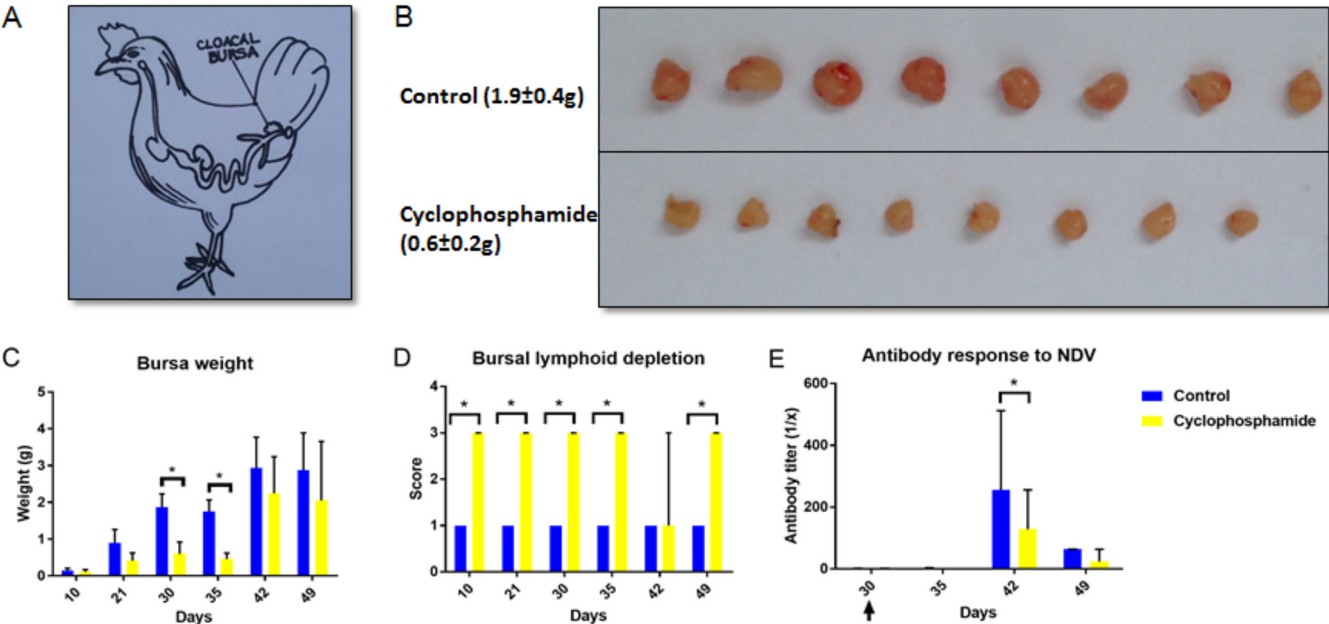

**FIG 2** Cyclophosphamide treatment reduced bursal weight and caused lymphoid depletion. (A) Anatomical localization of the cloacal bursa in birds. (B) Representative images of cloacal bursae at 35 days. Weights and standard deviations are shown, $n = 8$. The figure was spliced and pasted together to remove labels on the original photographs. (C) Bursal weight throughout the experiment. Data are representative of 8 samples/group (average ± SD). (D) Lymphoid depletion scores of the bursa, assessed by histopathology. Data are representative of 8 samples/group (median ± SD). (E) Antibody response to NDV, assessed by ELISA. The arrow indicates the day of vaccination. *, $P < 0.05$ by Kruskal Wallis test.

treated with cyclophosphamide (Fig. 2A to C). Added to this result, fecal IgA levels were significantly decreased in the chickens treated with cyclophosphamide (Fig. 3). Histopathological analysis showed marked lymphocyte depletion in the bursa (Fig. 2D). The effect of treatment also translated into lower systemic antibody responses to Newcastle disease virus (NDV) vaccination at 42 days of age (Fig. 2E).

To evaluate the effect of administration of cyclophosphamide on the cecal microbiota, the V4 variable region of the 16S rRNA gene was sequenced on an Illumina Miseq sequencer using birds 35 days of age, when the effect of cyclophosphamide on immune cells was evident. The cecal contents of 4 birds from each treatment were collected, and the metagenomic DNA was purified and used as the template for 16S rRNA gene amplification. The number of reads ranged from 12,671 to 143,650 in the libraries. For further analyses, the reads of each library were scaled to the smallest library size of 12,671 reads. Principal-component analysis of the libraries showed a

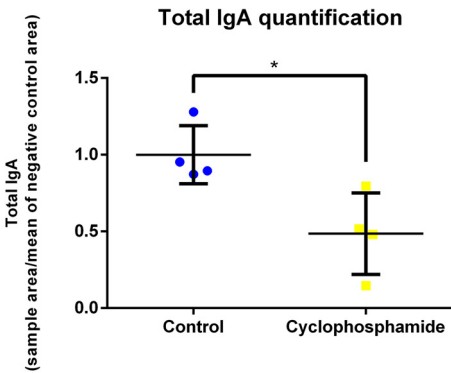

**FIG 3** Cyclophosphamide decreased total IgA levels in feces. *, $P < 0.05$ by the Wilcoxon test. Plot represents means and standard errors, $n = 4$.

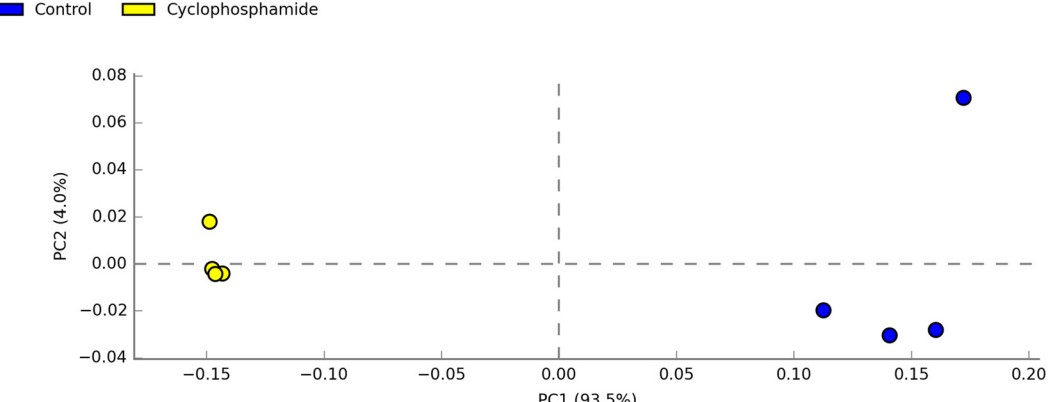

**FIG 4** Cyclophosphamide induced major alterations in microbiota composition. Comparison of cecal microbiota compositions between treatments. Principal-component analysis (PCA) of microbiota community by Bray-Curtis distance. PC1 is likely to represent the effect of the treatment over microbial populations.

distinct separation according to treatment and that the treatment (principal component 1 [PC1]) explained 93.5% of the differences between treatments (Fig. 4).

In addition, cyclophosphamide shifted the composition of cecal microbiota. Overall, 15 bacterial phyla (see Table S2) were identified by searching against the SILVA database, release 132. Among these phyla, *Firmicutes* was the most abundant in both treatments, accounting for, on average, 93.7% of all cecal bacterial sequences. At the genus level, 256 bacterial genera were identified using the SILVA database, release 132. Five genera (*Firmicutes* phyla) were statistically different in abundance between treatments by Welch's $t$ test ($P < 0.05$) and corrected with Bonferroni's test (Fig. 5). Among these five bacterial genera, three were more abundant in the cyclophosphamide treatment (*Lactobacillus*, *Blautia*, and *Faecalibacterium*), while two were more abundant in the control treatment (*Enterococcus* and *Weissella*) (Fig. 5). Besides that, a rarefaction plot showed an indication of how completely the cecal communities were covered by the sequencing (see Fig. S2). In addition, the administration of cyclophosphamide significantly ($P < 0.05$) increased the richness of the cecal community, indicated by alpha diversity metrics, such as observed operational taxonomic units (OTUs) and Shannon and Chao1 indices (Fig. 6).

Pearson's correlation was also used to evaluate the relationship between immune cells and microbiota. Pearson's correlation revealed significant correlations ($P < 0.05$) between treatments and microbiota. In the control treatment, there was no significant positive or negative relationship between immune cells and microbiota (Fig. 7A). Interestingly, in the cyclophosphamide treatment, there were significant negative correlations between B lymphocytes and the prevalence of *Lactobacillus* and *Faecalibacterium*, between CD45$^+$ CD8$^+$ cells and *Blautia*, and between monocytes and the prevalence of *Lactobacillus*, *Enterococcus*, and *Faecalibacterium* (Fig. 7B).

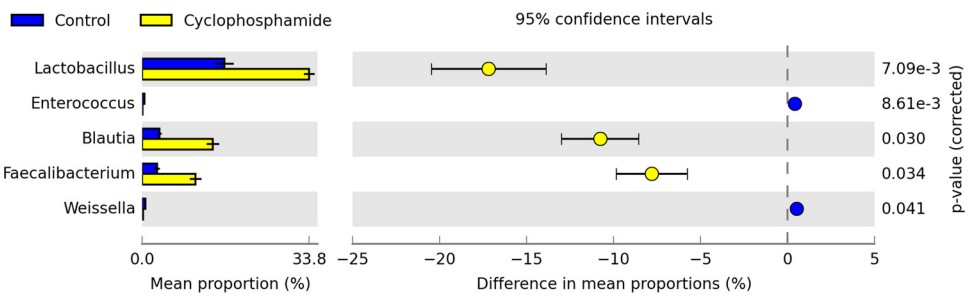

**FIG 5** Abundance of five bacterial genera was altered by cyclophosphamide. Statistical differences between treatments by Welch's $t$ test ($P < 0.05$) and corrected with Bonferroni's test. Data represent mean proportions (in %) and standard deviations per treatment, $n = 4$.

**A    Shannon index**

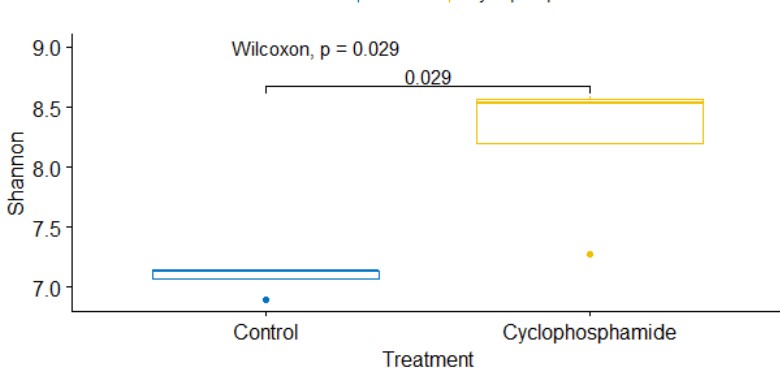

**B    Chao1 index**

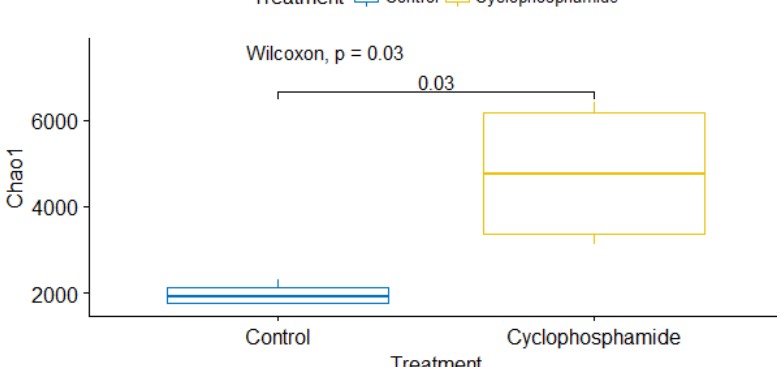

**C    OTUs number**

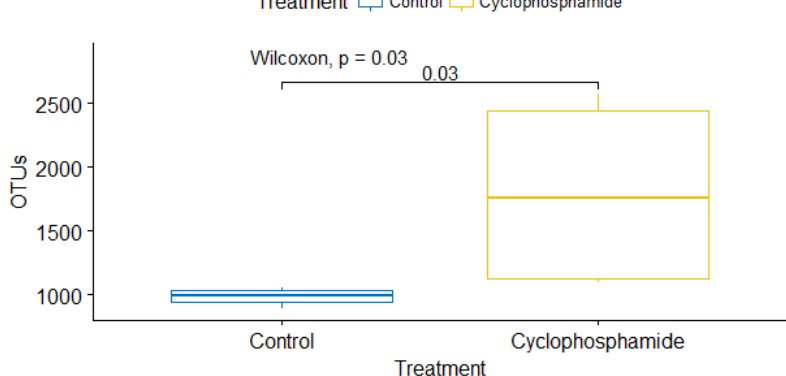

**FIG 6** Cyclophosphamide treatment increased cecal microbiota richness in different indexes, including the Shannon index (A), Chao1 index (B), and OTU number (C). *, $P < 0.05$ by the Wilcoxon test. Graphs represents means and standard errors, $n = 4$.

Additionally, to represent the behavior of the variables in a multivariate system, we used a nonmetric multidimensional scaling (NMDS) plot. This analysis represents the pairwise dissimilarity between objects in two-dimensional space, in this case, the cecal microbiota and the immune cells (plotted as vectors). NMDS results confirmed the clustering of the treatments in two groups with different bacterial genera compositions: the first, the control treatment, with higher abundance of *Enterococcus* and *Weissella*, and the second, the cyclophosphamide treatment, with higher abundance of *Lactoba-*

mSystems®

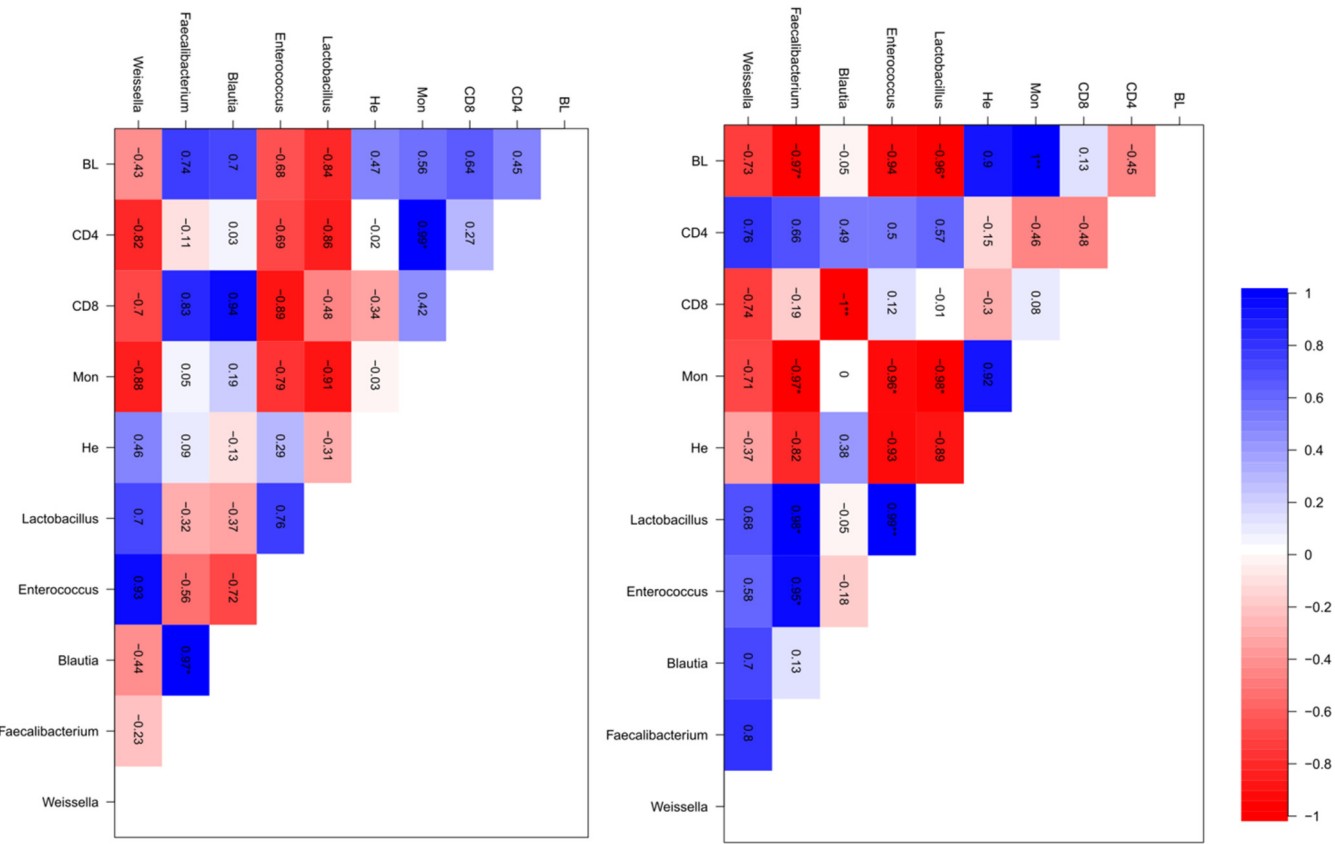

**FIG 7** Several bacterial genera correlated negatively with immune cell counts in the cyclophosphamide group. Correlation matrix showing Pearson's correlation coefficients between cecal bacterial genera and immune cells in the control group (A) or in the cyclophosphamide-treated group (B). Blue indicates a positive and red indicates a negative correlation (see bar on the right). The $R$ values are shown for each correlation inside the boxes. *, $P < 0.05$, $n = 4$. BL, B lymphocytes; CD4 and CD8, CD45$^+$ CD4$^+$ and CD45$^+$ CD8$^+$ cells; Mon, monocytes; He, heterophils.

*cillus*, *Blautia*, and *Faecalibacterium* (Fig. 8). In addition, the vectors representing the immune cells were in the opposite direction to the cyclophosphamide treatment, showing an inverse correlation of these parameters with bacteria present in high quantities in this treatment (Fig. 8), as seen in Pearson's correlation (Fig. 7B).

## DISCUSSION

Bacterial genera found in the present study are in overall agreement with previous reports of the intestinal microbiota in chickens (8, 9). The cyclophosphamide treatment, however, led to significant changes in the cecal microbiota, which correlated with the induced immunosuppression. The bacterial genera increased in the cecum by the cyclophosphamide treatment are generally associated with beneficial effects. For example, *Blautia* is an important producer of propionate, a central source of energy for enterocytes of cecum and ileum. *Lactobacillus* produces lactic acid, and its beneficial effects for poultry have been widely reported, including higher productivity (10, 11). Likewise, in the intestine, *Faecalibacterium* is a major producer of butyrate (12), another important short-chain fatty acid with beneficial effects to the host.

Mice treated intraperitoneally with cyclophosphamide also had significant shifts in the intestinal microbiota. For example, Xu and Zhang (13) observed that the families *Lachnospiraceae*, *Lactobacillaceae*, and *Staphylococcaceae* were significantly more abundant in fecal samples of mice treated with cyclophosphamide than in the control group by using massive 16S rRNA gene sequencing. These results agree with ours, since the genus *Blautia* belongs to the *Lachnospiraceae* family and the genus *Lactobacillus* belongs to the *Lactobacillaceae* family. It is possible that the microbiota shifts observed in the present work were due to NDV vaccination. There is evidence that NDV infection

## Relationships between immune cells and microbiota

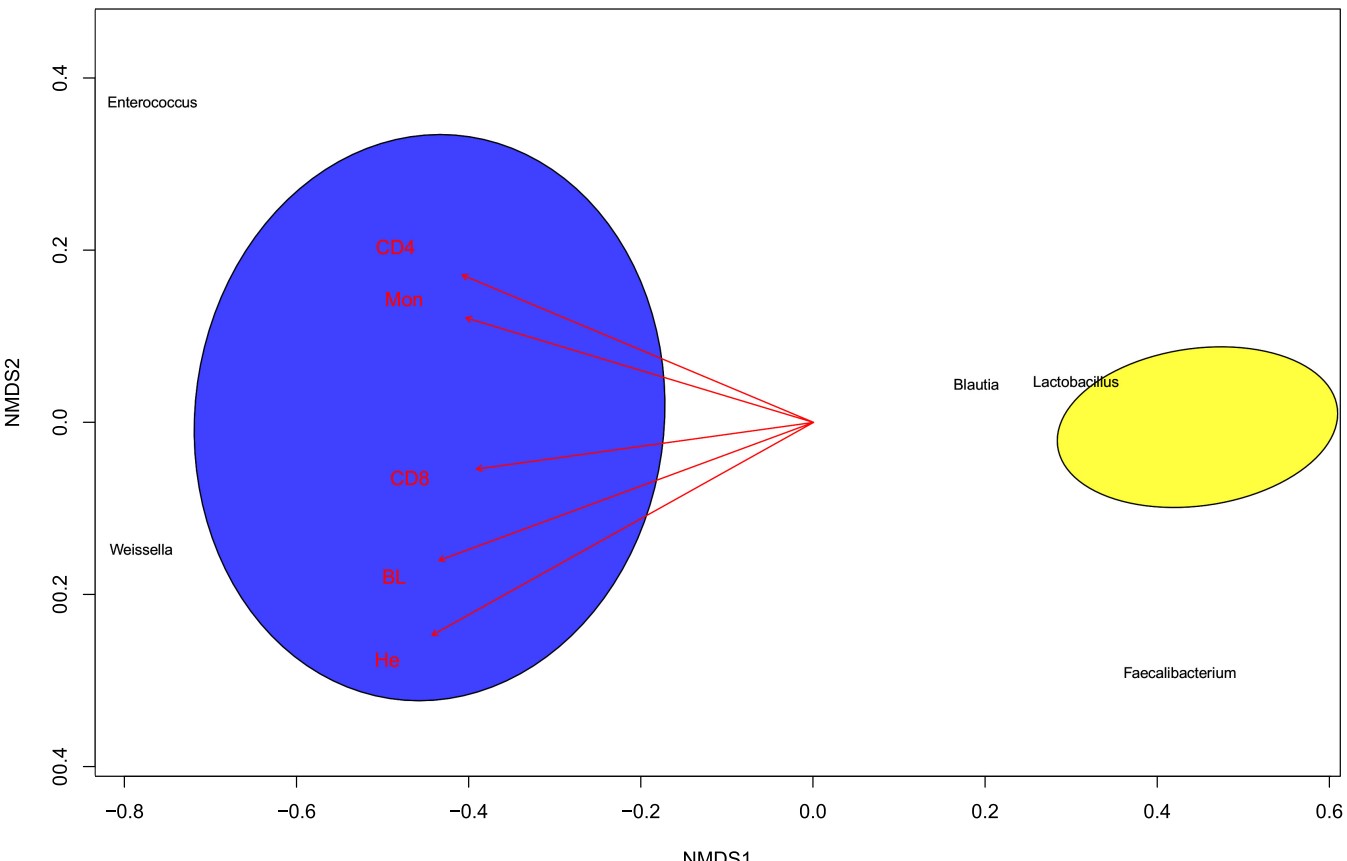

**FIG 8** Nonmetric multidimensional scaling (NMDS) plot. The ellipses encompass different treatments, the control group in blue and the cyclophosphamide group in yellow. Bacterial genera are show in black and immune cells in red vectors. The direction of the vectors indicates an inverse relationship with bacteria present in high quantities in the cyclophosphamide group.

interferes with the formation of intestinal microbiota in newly hatched chicks (14). Notwithstanding, in this study, the vaccination was given only at 30 days posthatch, when the intestinal microbiota is more stable.

In this study, we used four replicates by treatment; other studies have used similar sample sizes for microbiota assays. For example, Handl et al. (15) used four fecal samples per treatment; Videnska et al. (16) and Polansky et al. (17) employed three fecal samples per treatment for each collection point. Nevertheless, some tools are available to conduct calculations of power and sample sizes for microbiota assays; for example, La Rosa et al. (18) showed that the Dirichlet-multinomial distribution can be used to calculate power and sample sizes for experimental design and to perform tests of hypotheses (e.g., compare microbiomes across groups). The data from the saliva microbiota of this work showed that with a significance level of 5%, number of subjects at 10, and number of reads per subject at 10,000, the study has 53% power to detect the effect observed in the data. Other tools to test for differences in bacterial taxon compositions between groups of metagenomic samples are multivariate nonparametric methods based on permutation test such as analysis of similarity (ANOSIM), nonparametric (NP) Manova, and Mantel tests are widely used among community ecologists for this purpose. In our work, the Mantel test was applied with a total number of samples set at 8 before the NMDS analyses were executed. Following this protocol, without adequate value in the Mantel test, the NMDS analysis could not proceed. This shows that the "*n*" of the present work had the capacity to show the differences between the treatments. In addition, the principal-component analysis (PCA) based on

the Bray Curtis method shows the dissimilarity among the samples. The results in our work show the samples clustered clearly according the treatments, indicating that the number of samples analyzed was adequate.

The changes observed in the compositions of the chicken cecal microbiota may be due to an indirect effect of immune suppression by cyclophosphamide. Under normal conditions, intestinal B lymphocytes are activated and differentiated in Peyer's patches (19). B lymphocytes migrate to the intestinal lamina propria and secrete IgA, which is transported through the intestinal epithelium into the lumen (20). There, IgA protects the mucosal epithelium, preventing invasion by pathogens (21). This antibody isotype also modulates the composition of the intestinal microbiota (22) by binding to commensal bacteria in the intestinal lumen and enabling their transport to the lamina propria, where the bacteria interact with phagocytes (23). This process of luminal sampling of commensal bacteria by the immune system induces tolerance to the intestinal microbiota while also keeping it in check. For instance, mucosal antibodies generated against *Proteobacteria* alter the maturation of the entire microbiota (24).

Immunosuppression with cyclophosphamide led to significant decreases of several circulating leukocyte subsets in chicken. The data suggest changes in local immune responses may have affected the luminal sampling and thus the cecal microbiota. Counts of circulating B lymphocytes and $CD45^+$ $CD4^+$ cells were diminished in immunosuppressed animals, leading to decreased serum antibody production, as seen by a lower systemic antibody response to NDV. Also, $CD45^+$ $CD4^+$ TCR $V\beta1^+$ cells have crucial helper function in the intestines for IgA production in chickens (25). These cells were also suppressed in blood following cyclophosphamide administration. In addition, local IgA production was also decreased, which could lead to alterations in the microbiota (26, 27).

In addition, phagocytic cells were also impaired by cyclophosphamide. A reduced number of phagocytic and $CD45^+$ $CD4^+$ cells could lead to a decrease in the immune sampling in the intestinal lumen, thus possibly leading to a disruption of the microbial balance (Fig. 9). Supporting this interpretation, the Pearson's correlation for the cyclophosphamide treatment showed a significant negative correlation between B lymphocytes and *Lactobacillus*. Additionally, cecal biodiversity increased in birds treated with cyclophosphamide, indicating that a reduction in luminal sampling may have allowed expansion of certain bacterial genera in the intestine.

Increase in *Lactobacillus* may also function as a compensation mechanism. In normal chickens, defense mechanisms such as innate immunity, mechanical mucosal barrier, and colonization resistance prevent the translocation of endogenous bacteria from the intestine (28). Intestinal colonization resistance is the resistance to colonization by ingested bacteria or inhibition of overgrowth of resident bacteria normally present at low levels within the intestinal tract (28). This mechanism is mainly maintained by the beneficial effects of predominantly anaerobic resident microbiota (e.g., *Lactobacillus*) in the intestine, which, by their sheer numbers, provide resistance toward invasion (29). Thus, the increase of *Lactobacillus* observed in this work could suggest a possible mechanism of colonization resistance that promotes health and counteracts the deleterious effects of cyclophosphamide in the intestine.

Indeed, Nakamoto et al. (30) observed an enrichment of *Lactobacillus* in the guts of mice treated with concanavalin A, a potent immunosuppressant at high doses (31). These authors also observed an increase of intestinal permeability in treated animals. It is possible to speculate that the interaction between the immune system and *Lactobacillus* may be conserved in chickens and mice. Additionally, *Lactobacillus* has been shown to activate IL-22 production by gut lymphoid cells decreasing intestinal permeability and improving mucosal barrier function (30).

On the other hand, direct mechanisms of cyclophosphamide could also play a role in the intestinal microbiota. Selective suppression of intestinal microbiota by antitumor drugs has been proposed (32). The direct influence of antineoplastic drugs on bacterial viability was previously assessed and few drugs showed appreciable antibacterial activities (33). The possibility remains that antitumor drugs possess synergistic antimi-

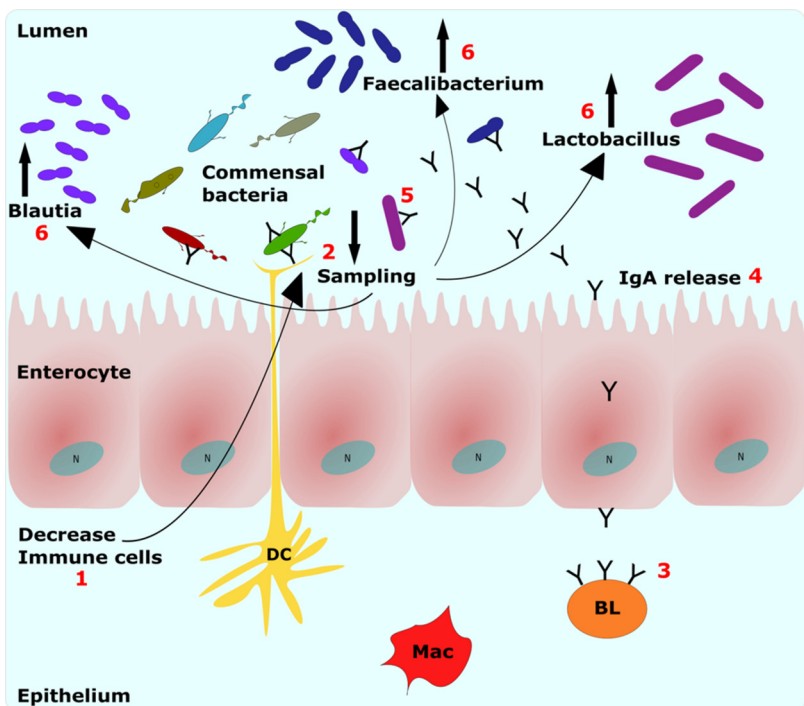

**FIG 9** Proposed mechanism for the changes in the microbiota following cyclophosphamide treatment. Step 1, decrease in immune cells in the epithelium leads to step 2, decrease in sampling of luminal contents by phagocytes. As consequence, step 3, there is a decrease in the number of B lymphocytes in the lamina propria and lower antibody production that leads to step 4, lower IgA release in the lumen by enterocytes. Thus, there is looser control of commensal bacteria by IgA, leading to step 5. In this way, the luminal sampling also is decreased, leading to step 6, disruption of intestinal balance; thus, some predominant groups such as *Lactobacillus* grow due to loss of control by the immune system. BL, B lymphocytes; Mac, macrophages; DC, dendritic cell; IgA, immunoglobulin A; N, nucleus.

crobial activity in combination with each other or with antibacterial therapeutic agents (34). In addition, other potential concurrent mechanisms, such as quorum sensing within the microbiota, also play a critical role in shaping the intestinal microbiota (35).

In conclusion, the administration of cyclophosphamide affected the overall composition of the cecal microbiota, causing an increase in concentrations of *Lactobacillus*, *Blautia*, and *Faecalibacterium*. These effects are possibly due to direct and indirect mechanisms. It remains to be determined if the enriched genera can lead to variations of metabolic profiles associated with potential beneficial effects to the host. Finally, it is interesting to observe how a systemically administered drug elicited shifts in the local intestinal microbiota. The detailed mechanisms that lead to these changes in the microbiota need further investigation.

## MATERIALS AND METHODS

**Experimental design and sampling.** Eighty male newborn chicks (Cobb) were divided in two groups (40 birds per group): control and cyclophosphamide treated. To the latter, cyclophosphamide (3 mg/kg) was administered subcutaneously during the first 4 days of life, according to previously published guidelines for inducing immunosuppression in chickens (36). Water and feed were given *ad libitum* throughout the experiment. Birds were housed for 49 days in isolators with HEPA-filtered airflow.

At 10, 21, 35, 42, and 49 days of age, eight blood samples per treatment were collected for immune cell measurement by flow cytometry (total leukocytes, B lymphocytes, CD45+ CD4+ and CD45+ CD8+ cells, and phagocytic cells [monocytes and heterophils]). CD45+ CD4+ TCR Vβ1+ cells were assessed from 35 days of age.

Antibody responses were assessed by immunization of chickens with an inactivated Newcastle disease vaccine (Ceva) at 30 days of age. Hemagglutination inhibition was used to assess antibody titers against Newcastle disease virus (NDV) at 30, 35, 42, and 49 days of age. At 10, 21, 30, 35, 42, and 49 days of age, the cloacal bursae were extracted and weighed from eight birds per treatment following euthanasia. Bursal lesions were assessed by histopathology at these time points. Additionally, samples of cecal contents were collected for DNA extraction, purification, and sequencing. Two isolators were used; 4 birds sampled per isolator at 35 days for cecal content were taken from a single isolator for treatment.

Immediately after euthanasia, the abdominal cavity was exposed, and the cecum was dissected from the other intestinal sections. Cecal contents were collected in sterile 2-ml tubes, stored on ice, and later stored at −20°C until processing. In total, eight samples of cecal contents were collected (four chickens per treatment at 35 days of age).

All animal procedures were approved by the Animal Experimentation Ethics Committee of the Federal University of Paraná (authorization CEUA-Bio UFPR 018/2017).

**Flow cytometry.** Whole-blood samples were processed according to a no-lyse-no-wash modified protocol from Stabel et al. (37). Briefly, 50 $\mu$l of whole blood was incubated for 30 min at 37°C with antibodies against one of the following chicken cellular markers: CD4, CD8, TCR V$\beta$1, or Bu-1 (Southern Biotechnology). All samples were also stained for CD45, used for gating the leukocyte population (37). Samples were fixed with 1% paraformaldehyde for 30 min at 4°C. Subsequently, the samples were diluted with phosphate-buffered saline (PBS) to a final volume of 2 ml. Phagocytic cells and their activity were assessed using the pHrodo fluorescein isothiocyanate (FITC) reagent (Thermo Scientific). Fifty microliters of pHrodo was added to 50 $\mu$l of whole blood for 30 min at 37°C. Cells were then stained for CD45 as described in reference 37, and cells were assessed for the number of green fluorescence events (phagocytic cells) and the intensity of fluorescence (thus inferring phagocytic activity). Phagocytic cells were further discriminated regarding cell granularity (side scatter). High-granularity fluorescent cells are here named phagocytic heterophils, and low-granularity fluorescent cells are referred to as phagocytic monocytes.

For absolute quantification of leukocytes, CountBright beads (Thermo Scientific) were added to the tube. Samples were read in a FACSCalibur flow cytometer (Becton and Dickinson) equipped with an argon laser. More detailed methods regarding the flow cytometry experiments are shown in Fig. S1 and Text S2 in the supplemental material.

**NDV hemagglutination inhibition and bursal histopathology.** Antibody responses to NDV were assessed by enzyme-linked immunosorbent assay (ELISA), using commercial kits (Idexx Laboratories). Samples of the bursae were processed routinely for histology and stained with hematoxylin and eosin. Twenty fields per treatment were scanned under a light microscope (Olympus BX41 Olympus USA) at ×40 magnification. Bursal histopathology analyses were performed by a trained veterinarian through the subjective conventional method. The fields were classified into depletion scores that varied from 1 to 4 (score 1, depletion of <25%; score 2, depletion of 25% to 50%; score 3, depletion of 50% to 75%; score 4, depletion of >75%).

**DNA extraction, 16S rRNA gene amplification, and sequencing.** Genomic DNA from each sample was isolated from 150 mg of cecal luminal content using the ZR Fecal DNA MiniPrep kit (Zymo Research, Inc.). The variable V4 region of 16S rRNA gene was amplified using the universal primers 515F and 806R (38) and KlenTaq Master Mix (Sigma); amplification controls without template were employed. The PCR conditions used were 94°C for 3 min, 18 cycles of 94°C for 45 s, 50°C for 30 s, and 68°C for 60 s, followed by 72°C for 10 min. The amplicons were quantified with Qubit using an HS dsDNA kit (Invitrogen), diluted to 500 pM, and pooled. Then, 16 pM of pooled DNA was sequenced using MiSeq reagent 500V2. Sequencing was performed using an Illumina MiSeq sequencer (Illumina) obtaining paired-end reads of 250 bp as described previously (38).

**Diversity analysis.** Sequencing data were analyzed with the QIIME pipeline (39). The read outputs were resampled to 12,671 per sample, allowing for direct diversity comparison. A summary with the reads before and after normalization is shown in Table S1. Sequences were filtered by quality and classified into bacterial genera through the recognition of operational taxonomic units (OTUs) based on the identity at 97% of the sequences compared to the SILVA 132 ribosomal sequence database, 2018 release (40, 41). The file table_even.biom containing the OTUs identified in QIIME was converted to spf format using the script biom_to_stamp.py available through the Microbiome Helper website (https://github.com/mlangill/microbiome_helper/wiki) (42). It was transferred to the statistical program STAMP (Statistical Analysis of Metagenomic Profiles) (43) for basic diversity analyses of the proportions of sequences per genus and PCA plotting.

**Total fecal IgA levels.** Two grams of fresh fecal sample was collected directly from the rectums of euthanized animals. Stool samples were analyzed by Bradford method for total protein quantification. A standard amount of protein (19.78 $\mu$g) was then run on an SDS-PAGE gel for electrophoretic separation of molecules. The gel was used for Western blotting. IgA was detected with a horseradish peroxidase (HRP)-coupled polyclonal specific antibody at 1:5,000 (Bio Rad, AAI28P). The result was developed on a radiographic film. Film exposure was nonsaturating, that is, the exposure of the radiographic film was controlled to allow the relative quantification of IgA. Bands corresponding to IgA were quantified by densitometry with the software ImageJ (44). The amount of IgA was evaluated in relation to the total amount of protein of each sample, obtaining the amount of total IgA in each sample of feces.

**Statistical analysis.** Welch's $t$ tests ($P < 0.05$) corrected with Bonferroni's tests were used to compare bacterial genera abundance between treatments. Both analyses were included in the STAMP software. Biodiversity richness (Shannon index, Chao1 index, and OTU number) and total IgA levels in feces were compared with Wilcoxon's tests ($P < 0.05$). Immune cell types and numbers were compared with the Kruskal Wallis test ($P < 0.05$). Bursal weight, lymphoid depletion, and antibody response to NDV were also analyzed by Kruskal Wallis tests ($P < 0.05$). Pearson's correlation analysis and nonmetric multidimensional scaling (NMDS) were performed using the psych and vegan packages included in R software (45). Only statistically significant results are reported ($P < 0.05$).

**Data availability.** The data set was submitted to the NCBI site under the BioSample accession code SAMN08457450. In addition, the R scripts used for Pearson and NMDS analyses are available in Text S1 in the supplemental material.

## SUPPLEMENTAL MATERIAL

Supplemental material is available online only.

**TEXT S1**, DOCX file, 0.1 MB.

**TEXT S2**, DOCX file, 0.1 MB.

**FIG S1**, DOCX file, 0.7 MB.

**FIG S2**, DOCX file, 0.3 MB.

**FIG S3**, DOCX file, 0.1 MB.

**TABLE S1**, DOCX file, 0.1 MB.

**TABLE S2**, DOCX file, 0.1 MB.

## ACKNOWLEDGMENTS

We thank Roseli Prado, Fernanda Rigo, and Marilza Doroti Lamour for technical assistance, David Mitchell for critical reading of the manuscript, and the artist Eliane Leite for ceding the chicken figure.

This work was supported by the National Institute of Science and Technology of Biological Nitrogen Fixation/CNPq/MCT and Araucária Foundation.

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
