## [Reviewer comments · mSystems]

Cyclophosphamide increases *Lactobacillus* in the intestinal microbiota in chickens

Dany Mesa, Breno Beirão, Eduardo Balsanelli, Luiz Sesti, Luiz Caron, Leonardo Cruz, and Emanuel Maltempi de Souza

Corresponding Author(s): Emanuel Maltempi de Souza, Federal University of Parana

Review Timeline:

Submission Date:	January 30, 2020
Editorial Decision:	February 25, 2020
Revision Received:	March 20, 2020
Editorial Decision:	May 26, 2020
Revision Received:	June 17, 2020
Accepted:	June 27, 2020

Editor: Robert Beiko

Reviewer(s): The reviewers have opted to remain anonymous.

Transaction Report:

DOI: <https://doi.org/10.1128/mSystems.00080-20>

February 25, 2020

Mr. Dany Mesa
Federal University of Paraná
Biochemistry and Molecular Biology
Curitiba, Paraná 82900-420
Brazil

Re: mSystems00080-20 (Cyclophosphamide increases *Lactobacillus* in the intestinal microbiota in chickens)

Dear Mr. Dany Mesa:

I now have two reviews of your revised manuscript in hand. I am pleased to accept the manuscript in principle, pending minor modifications. Please address all referees' comments in your revised submission.

Below you will find the comments of the reviewers.

To submit your modified manuscript, log onto the eJP submission site at <https://msystems.msubmit.net/cgi-bin/main.plex>. If you cannot remember your password, click the "Can't remember your password?" link and follow the instructions on the screen. Go to Author Tasks and click the appropriate manuscript title to begin the resubmission process. The information that you entered when you first submitted the paper will be displayed. Please update the information as necessary. Provide (1) point-by-point responses to the issues raised by the reviewers as file type "Response to Reviewers," not in your cover letter, and (2) a PDF file that indicates the changes from the original submission (by highlighting or underlining the changes) as file type "Marked Up Manuscript - For Review Only."

Please return the manuscript within 60 days; if you cannot complete the modification within this time period, please contact me. If you do not wish to modify the manuscript and prefer to submit it to another journal, please notify me of your decision immediately so that the manuscript may be formally withdrawn from consideration by mSystems.

To avoid unnecessary delay in publication should your modified manuscript be accepted, it is important that all elements you upload meet the technical requirements for production. I strongly recommend that you check your digital images using the Rapid Inspector tool at <http://rapidinspector.cadmus.com/RapidInspector/zmw/>.

Sincerely,

Robert Beiko

Editor, mSystems

Journals Department
Reviewer comments:

Reviewer #1 (Comments for the Author):

This manuscript, which I have reviewed before, describes the effect of cyclophosphamide-induced immunosuppression on the cecal microbiota of broiler chickens. The authors have revised their manuscript according to the comments provided in the first round of review. Overall the authors have done a good job of addressing the comments I made in the previous round, with a couple of small exceptions that I have outlined below.

Specific comments:

Methods/Results

1. Methods: Regarding the statistical power of the chosen number of replicates: The authors have provided a fine explanation in their rebuttal, but as far as I can tell have made no changes to the manuscript that readers will encounter to indicate this. Please consider including some of this text and references in the revised manuscript so that readers can be assured of the statistical power of the methods employed by the authors.

5. Regarding sampling effort: The newly included Table S1 reasonably provides the number of reads observed in each sample along with the downsampled/normalized numbers, but does not provide an indication of how completely the cecal communities were covered at this sequencing effort. Please consider including some metric of this such as rarefaction or Good's coverage estimator so that readers can assess the possible effects on the alpha diversity metrics employed by the authors, even if it is acknowledged that any biases are equally applied across the samples analyzed in this particular study.

Reviewer #2 (Comments for the Author):

The re-submission of the work by Mesa et al demonstrates the immune effects of cyclophosphamide in chickens and alterations of the cecal microbiome. The authors have addressed several of the prior reviewer comments, but there are a few additional critiques to offer

(line items derived from the "marked up" version of the manuscript):

1. Line 25-27: This line still reads awkwardly and grammatically incorrect despite revision.
2. Line 65: The word "of" is not needed.
3. Line 68: The author should specify that the treatment translated into lower systemic antibody responses.
4. Line 135-146: The author explains how changes in B lymphocytes and IgA production can indirectly alter gut microbiota. Later, the author notes (Line 182-183) that direct mechanisms could also play a role. It would be useful to include commentary on these potential direct effects and other potential concurrent mechanisms, such as quorum sensing within the microbiome.
5. Line 150: The author should specify B lymphocytes as systemic.
6. Line 152: Grammatically, "be" should be "was."
7. Line 180: Grammatically, this sentence should read "In conclusion,".
8. For flow cytometric methods:

The author indicates that fluorochrome conjugate information was added; however, it is not included in for all antibodies used. A useful guide for appropriate flow cytometric methods descriptions and T cell staining in chickens may be found in a recent article: Dai, M.; Li, S.; Shi, K.; Liao, J.; Sun, H.; Liao, M. Systematic Identification of Host Immune Key Factors Influencing Viral Infection in PBL of ALV-J Infected SPF Chicken. *Viruses* 2020, 12, 114. I recommend adding the fluorochrome information for all antibodies and additionally providing the specific cell surface protein-target for the B lymphocyte antibody.

A representative flow gating strategy was included, but it only provides gating strategies for CD45+CD4+ cells and phagocytes. Dot plot gating strategies should be included for all cell types analyzed (B lymphocytes, CD4+ cells, CD8+ cells, and phagocytes). Additionally, missing from the gating strategy is doublet discrimination, which should be done in all analyses.

It is still inappropriate to refer to CD45+CD4+ cells and CD45+CD8+ as T lymphocytes, despite the author's rationale given in the response to reviewer comments. Without CD3 or other discriminating markers on the specific cells in the analyses, these cells can only be referred to by the identified markers; and the lack of CD3 labeling is important enough that the author should make this explicitly clear in the body of the paper. CD45+CD4+ cells can include T lymphocytes, NK cells, monocytes, and granulocytes. Altered gating based on size and granularity of the CD45 population and additional staining markers are needed to properly interrogate these populations. The additional triple labeling experimental data trends are compelling. However, the day 35 data results of the triple labeled experiments do offer different insights than Figure 1C: the mean number of CD45+CD4+ cells from control chickens appears to be more than the number of CD4+TCRV β 1+ cells from the same group resulting in significant results at day 35 in the former and not the latter. This could be due to experimental variability or alterations in the aforementioned CD4+ cell populations. I recommend including the triple stained cell experiments within paper in order to further support the conclusion that T lymphocyte populations are altered with cyclophosphamide treatment.

Lastly, it is important to note that in the evaluation of T cell subsets, typical flow cytometric gating strategies evaluate CD4 and CD8 labeling on a two-parameter density plot to delineate CD4+CD8-, CD4+CD8+, and CD4-CD8+ cells. This cannot be done without dual labeling but should be considered in future experiments.

9. Line 238: It is presumed that a board-certified veterinary pathologist was utilized and not just a trained veterinarian. If that is the case, please indicate as such.
10. Line 269: As with the antibodies used for flow cytometric methods, I recommend adding more detailed antibody information
11. For total IgA experiments, there is no description of how the fecal samples were collected. Please include this information. Additionally, the figure legend for this data (Figure 3) does not provide adequate information: it should include (as the other figure legends do) the total N, what

data is represented (mean versus median \pm SD or SEM), and what the asterisk represents.

The re-submission of the work by Mesa *et al* demonstrates the immune effects of cyclophosphamide in chickens and alterations of the cecal microbiome. The authors have addressed several of the prior reviewer comments, but there are a few additional critiques to offer (line items derived from the “marked up” version of the manuscript):

1. Line 25-27: This line still reads awkwardly and grammatically incorrect despite revision.
2. Line 65: The word “of” is not needed.
3. Line 68: The author should specify that the treatment translated into lower systemic antibody responses.
4. Line 135-146: The author explains how changes in B lymphocytes and IgA production can indirectly alter gut microbiota. Later, the author notes (Line 182-183) that direct mechanisms could also play a role. It would be useful to include commentary on these potential direct effects and other potential concurrent mechanisms, such as quorum sensing within the microbiome.
5. Line 150: The author should specify B lymphocytes as systemic.
6. Line 152: Grammatically, “be” should be “was.”
7. Line 180: Grammatically, this sentence should read “In conclusion,”.
8. For flow cytometric methods:

The author indicates that fluorochrome conjugate information was added; however, it is not included in for all antibodies used. A useful guide for appropriate flow cytometric methods descriptions and T cell staining in chickens may be found in a recent article: Dai, M.; Li, S.; Shi, K.; Liao, J.; Sun, H.; Liao, M. Systematic Identification of Host Immune Key Factors Influencing Viral Infection in PBL of ALV-J Infected SPF Chicken. *Viruses* 2020, *12*, 114. I recommend adding the fluorochrome information for all antibodies and additionally providing the specific cell surface protein-target for the B lymphocyte antibody.

A representative flow gating strategy was included, but it only provides gating strategies for CD45⁺CD4⁺ cells and phagocytes. Dot plot gating strategies should be included for all cell types analyzed (B lymphocytes, CD4⁺ cells, CD8⁺ cells, and phagocytes). Additionally, missing from the gating strategy is doublet discrimination, which should be done in all analyses.

It is still inappropriate to refer to CD45⁺CD4⁺ cells and CD45⁺CD8⁺ as T lymphocytes, despite the author’s rationale given in the response to reviewer comments. Without CD3 or other discriminating markers on the specific cells in the analyses, these cells can only be referred to by the identified markers; and the lack of CD3 labeling is important enough that the author should make this explicitly clear in the body of the paper. CD45⁺CD4⁺ cells can include T lymphocytes, NK cells, monocytes, and granulocytes. Altered gating based on size and granularity of the CD45 population and additional staining markers are needed to properly interrogate these populations. The additional triple labeling experimental data trends are compelling. However, the day 35 data results of the triple labeled experiments do offer different insights than Figure 1C: the mean number of CD45⁺CD4⁺ cells from control chickens appears to be more than the number of CD4⁺TCRVβ1⁺ cells from the same group resulting in significant results at day 35 in the former and not the latter. This could be due to experimental variability or alterations in the aforementioned CD4⁺ cell populations. I recommend including the triple stained cell

experiments within paper in order to further support the conclusion that T lymphocyte populations are altered with cyclophosphamide treatment.

Lastly, it is important to note that in the evaluation of T cell subsets, typical flow cytometric gating strategies evaluate CD4 and CD8 labeling on a two-parameter density plot to delineate CD4⁺CD8⁻, CD4⁺CD8⁺, and CD4⁻CD8⁺ cells. This cannot be done without dual labeling but should be considered in future experiments.

9. Line 238: It is presumed that a board-certified veterinary pathologist was utilized and not just a trained veterinarian. If that is the case, please indicate as such.
10. Line 269: As with the antibodies used for flow cytometric methods, I recommend adding more detailed antibody information
11. For total IgA experiments, there is no description of how the fecal samples were collected. Please include this information. Additionally, the figure legend for this data (Figure 3) does not provide adequate information: it should include (as the other figure legends do) the total N, what data is represented (mean versus median +/- SD or SEM), and what the asterisk represents.

Comments to editor:

I still believe the content of this paper is interesting and the data contribute to the field in a significant fashion. I am, however, still concerned about the oversight of the authors with respect to the flow cytometric methods and data as currently presented. The additional triple labeled data shown as a response to the reviewers is compelling, and if shown in full, could support the conclusions in the paper regarding T lymphocytes and may negate the need for additional experiments. Rewording regarding the CD45⁺CD4⁺ and CD45⁺CD8⁺ cells will still be needed. Thus, if the editor agrees, I believe the paper could be accepted with current experiments if appropriate revisions are applied.

Reviewer 1.

Methods/Results

1. Methods: Regarding the statistical power of the chosen number of replicates: The authors have provided a fine explanation in their rebuttal, but as far as I can tell have made no changes to the manuscript that readers will encounter to indicate this. Please consider including some of this text and references in the revised manuscript so that readers can be assured of the statistical power of the methods employed by the authors.

A. This has been added.

5. Regarding sampling effort: The newly included Table S1 reasonably provides the number of reads observed in each sample along with the downsampled/normalized numbers, but does not provide an indication of how completely the cecal communities were covered at this sequencing effort. Please consider including some metric of this such as rarefaction or Good's coverage estimator so that readers can assess the possible effects on the alpha diversity metrics employed by the authors, even if it is acknowledged that any biases are equally applied across the samples analyzed in this particular study.

A. This has been added as Figure S2.

Reviewer 2.

The re-submission of the work by Mesa *et al* demonstrates the immune effects of cyclophosphamide in chickens and alterations of the cecal microbiome. The authors have addressed several of the prior reviewer comments, but there are a few additional critiques to offer (line items derived from the "marked up" version of the manuscript):

1. Line 25-27: This line still reads awkwardly and grammatically incorrect despite revision.

A. The sentences were reformulated.

2. Line 65: The word "of" is not needed.

A. We reformulated the sentence for clarity.

3. Line 68: The author should specify that the treatment translated into lower systemic antibody responses.

A. This was changed accordingly.

4. Line 135-146: The author explains how changes in B lymphocytes and IgA production can indirectly alter gut microbiota. Later, the author notes (Line 182-183) that direct mechanisms could also play a role. It would be useful to include commentary on these potential direct effects and other potential concurrent mechanisms, such as quorum sensing within the microbiome.

A. We added a paragraph with this topic, line 200-207

5. Line 150: The author should specify B lymphocytes as systemic.

A. This has been corrected.

6. Line 152: Grammatically, “be” should be “was.”

A. This has been corrected.

7. Line 180: Grammatically, this sentence should read “In conclusion,”.

A. This has been corrected.

8. For flow cytometric methods: The author indicates that fluorochrome conjugate information was added; however, it is not included in for all antibodies used. A useful guide for appropriate flow cytometric methods descriptions and T cell staining in chickens may be found in a recent article: Dai, M.; Li, S.; Shi, K.; Liao, J.; Sun, H.; Liao, M. Systematic Identification of Host Immune Key Factors Influencing Viral Infection in PBL of ALV-J Infected SPF Chicken. *Viruses* 2020, *12*, 114. I recommend adding the fluorochrome information for all antibodies and additionally providing the specific cell surface protein-target for the B lymphocyte antibody.

A. We agree that this could have been enriched. Data now are in the revised Supplementary methods. B cell antigen is “Bu-1” itself, which names the antibody. This has been added in the first occurrence of “B lymphocytes” in the results.

A representative flow gating strategy was included, but it only provides gating strategies for CD45⁺CD4⁺ cells and phagocytes. Dot plot gating strategies should be included for all cell types analyzed (B lymphocytes, CD4⁺ cells, CD8⁺ cells, and phagocytes). Additionally, missing from the gating strategy is doublet discrimination, which should be done in all analyses.

A. Other graphs were added to conform to the requirements.

It is still inappropriate to refer to CD45⁺CD4⁺ cells and CD45⁺CD8⁺ as T lymphocytes, despite the author’s rationale given in the response to reviewer comments. Without CD3 or other discriminating markers on the specific cells in the analyses, these cells can only be referred to by the identified markers; and the lack of CD3 labeling is important enough that the author should make this explicitly clear in the body of the paper. CD45⁺CD4⁺ cells can include T lymphocytes, NK cells, monocytes, and granulocytes. Altered gating based on size and granularity of the CD45 population and additional staining markers are needed to properly interrogate these populations.

A. To conform to the recommendations of the reviewer, cell subpopulation naming has been altered to identify their “CD” phenotype only, with the exception of B lymphocytes (instead of “T lymphocytes”, they are now identified in the paper as “CD4” or “CD8”, for instance).

The additional triple labeling experimental data trends are compelling. However, the day 35 data results of the triple labeled experiments do offer different insights than Figure 1C: the mean number of CD45+CD4+ cells from control chickens appears to be more than the number of CD4+TCRVβ1+ cells from the same group resulting in significant results at day 35 in the former and not the latter. This could be due to experimental variability or alterations in the aforementioned CD4+ cell populations. I recommend including the triple stained cell experiments within paper in order to further support the conclusion that T lymphocyte populations are altered with cyclophosphamide treatment.

A. We believe it is expected that CD45+CD4+ cells are greater in number than CD45+CD4+TCRVb1+ cells are a subset of the former. Just to make clear, both measurements (double and triple stains) were performed on samples from the same animals. The difference in statistical significance are also due to the presence of data from different time points – ANOVA considers all data, and not only a single time point, to determine significance.

Lastly, it is important to note that in the evaluation of T cell subsets, typical flow cytometric gating strategies evaluate CD4 and CD8 labeling on a two-parameter density plot to delineate CD4+CD8-, CD4+CD8+, and CD4-CD8+ cells. This cannot be done without dual labeling but should be considered in future experiments.

A. We thank you for your suggestion. This will be noted for future experiments.

9. Line 238: It is presumed that a board-certified veterinary pathologist was utilized and not just a trained veterinarian. If that is the case, please indicate as such.

A. In Brazil, board certification of veterinary specialists is never a requirement and is certainly not common, unlike in the US or UK. The veterinarian was trained in Poultry Pathology by a residency at a prestigious university.

10. Line 269: As with the antibodies used for flow cytometric methods, I recommend adding more detailed antibody information

A. This has been added.

11. For total IgA experiments, there is no description of how the fecal samples were collected. Please include this information. Additionally, the figure legend for this data (Figure 3) does not provide adequate information: it should include (as the other figure legends do) the total N, what data is represented (mean versus median +/- SD or SEM), and what the asterisk represents.

A. These have been reformed as suggested.

Comments to editor: I still believe the content of this paper is interesting and the data contribute to the field in a significant fashion. I am, however, still concerned about the oversight of the authors with respect to the flow cytometric methods and data as currently presented. The additional triple labeled data shown as a response to the reviewers is compelling, and if shown in full, could support the conclusions in the paper regarding T lymphocytes and may negate the

need for additional experiments. Rewording regarding the CD45+CD4+ and CD45+CD8+ cells will still be needed. Thus, if the editor agrees, I believe the paper could be accepted with current experiments if appropriate revisions are applied.

May 26, 2020

Prof. Emanuel Maltempi de Souza
Federal University of Parana
Biochemistry and Molecular Biology
Curitiba, Parana 81531980
Brazil

Re: mSystems00080-20R1 (Cyclophosphamide increases *Lactobacillus* in the intestinal microbiota in chickens)

Dear Prof. Emanuel Maltempi de Souza:

Below you will find the comments of the reviewers.

To submit your modified manuscript, log onto the eJP submission site at <https://msystems.msubmit.net/cgi-bin/main.plex>. If you cannot remember your password, click the "Can't remember your password?" link and follow the instructions on the screen. Go to Author Tasks and click the appropriate manuscript title to begin the resubmission process. The information that you entered when you first submitted the paper will be displayed. Please update the information as necessary. Provide (1) point-by-point responses to the issues raised by the reviewers as file type "Response to Reviewers," not in your cover letter, and (2) a PDF file that indicates the changes from the original submission (by highlighting or underlining the changes) as file type "Marked Up Manuscript - For Review Only."

Due to the SARS-CoV-2 pandemic, our typical 60 day deadline for revisions will not be applied. I hope that you will be able to submit a revised manuscript soon, but want to reassure you that the journal will be flexible in terms of timing, particularly if experimental revisions are needed. When you are ready to resubmit, please know that our staff and Editors are working remotely and handling submissions without delay. If you do not wish to modify the manuscript and prefer to submit it to another journal, please notify me of your decision immediately so that the manuscript may be formally withdrawn from consideration by mSystems.

To avoid unnecessary delay in publication should your modified manuscript be accepted, it is important that all elements you upload meet the technical requirements for production. I strongly recommend that you check your digital images using the Rapid Inspector tool at <http://rapidinspector.cadmus.com/RapidInspector/zmw/>.

Corresponding authors may join or renew ASM membership to obtain discounts on publication fees. Need to upgrade your membership level? Please contact Customer Service at

Service@asmusa.org.

Sincerely,

Robert Beiko

Editor, mSystems

Journals Department
Reviewer comments:

Reviewer #2 (Comments for the Author):

The re-submission of the work by Mesa et al demonstrates the immune effects of cyclophosphamide in chickens and alterations of the cecal microbiome. The authors have addressed several of the prior reviewer comments, but not all have been appropriately addressed (line items derived from the "marked up" version of the manuscript):

1. Line 226, 459, 493: While the author has altered the language around CD4 and CD8 labeled cells in parts of the manuscript, they are still referred to as T lymphocytes inappropriately and need to be corrected to be congruent with the rest of the paper. These can only be referred to as CD4 or CD8+ cells as previously indicated.
2. Line 245: The only place Bu-1 is noted as the specific marker to identify B lymphocytes is line 58 in the text. It should be referred to in the primary methods as well: B lymphocyte is a cell type not an antigen or antibody name. The supplemental methods give a more appropriate level of detail, but primary antibody delineations are important enough to be in primary methods as well. Please change B lymphocyte to Bu-1 in the primary methods.
3. Line 135-136: This should not be a question and the transition, "on the other hand," is inappropriate making this paragraph appear out of place.
4. Figure and Figure Legend S1: This figure includes the triple labeled cells. However, the corresponding triple labeled data from specific timepoints is not represented anywhere in the paper. Without corresponding data or context for its inclusion, it is not useful information. This figure still does not show B lymphocyte gating. Doublet discrimination is still missing and can be done with re-analysis of current data. Please see prior review.

The re-submission of the work by Mesa *et al* demonstrates the immune effects of cyclophosphamide in chickens and alterations of the cecal microbiome. The authors have addressed several of the prior reviewer comments, but not all have been appropriately addressed (line items derived from the “marked up” version of the manuscript):

1. Line 226, 459, 493: While the author has altered the language around CD4 and CD8 labeled cells in parts of the manuscript, they are still referred to as T lymphocytes inappropriately and need to be corrected to be congruent with the rest of the paper. These can only be referred to as CD4 or CD8⁺ cells as previously indicated.
2. Line 245: The only place Bu-1 is noted as the specific marker to identify B lymphocytes is line 58 in the text. It should be referred to in the primary methods as well: B lymphocyte is a cell type not an antigen or antibody name. The supplemental methods give a more appropriate level of detail, but primary antibody delineations are important enough to be in primary methods as well. Please change B lymphocyte to Bu-1 in the primary methods.
3. Line 135-136: This should not be a question and the transition, "on the other hand," is inappropriate making this paragraph appear out of place.
4. Figure and Figure Legend S1: This figure includes the triple labeled cells. However, the corresponding triple labeled data from specific timepoints is not represented anywhere in the paper. Without corresponding data or context for its inclusion, it is not useful information. This figure still does not show B lymphocyte gating. Doublet discrimination is still missing and can be done with re-analysis of current data. Please see prior review.

Comments to editor:

I still believe the content of this paper is interesting and the data contribute to the field in a significant fashion. Rewording regarding the CD45+CD4+ and CD45+CD8+ cells in all appropriate locations is still needed. Thus, if the editor agrees, I believe the paper could be accepted with current experiments if appropriate revisions are applied.

Answers

The re-submission of the work by Mesa *et al* demonstrates the immune effects of cyclophosphamide in chickens and alterations of the cecal microbiome. The authors have addressed several of the prior reviewer comments, but not all have been appropriately addressed (line items derived from the “marked up” version of the manuscript):

1. Line 226, 459, 493: While the author has altered the language around CD4 and CD8 labeled cells in parts of the manuscript, they are still referred to as T lymphocytes inappropriately and need to be corrected to be congruent with the rest of the paper. These can only be referred to as CD4 or CD8+ cells as previously indicated.

A. This was changed accordingly.

2. Line 245: The only place Bu-1 is noted as the specific marker to identify B lymphocytes is line 58 in the text. It should be referred to in the primary methods as well: B lymphocyte is a cell type not an antigen or antibody name. The supplemental methods give a more appropriate level of detail, but primary antibody delineations are important enough to be in primary methods as well. Please change B lymphocyte to Bu-1 in the primary methods.

A. This was changed accordingly.

3. Line 135-136: This should not be a question and the transition, "on the other hand," is inappropriate making this paragraph appear out of place.

A. The sentence was reformulated.

4. Figure and Figure Legend S1: This figure includes the triple labeled cells. However, the corresponding triple labeled data from specific timepoints is not represented anywhere in the paper. Without corresponding data or context for its inclusion, it is not useful information. This figure still does not show B lymphocyte gating. Doublet discrimination is still missing and can be done with re-analysis of current data. Please see prior review.

A. The data for CD45+CD4+TCRVb1+ triple labeled cells were included as requested in the original review, as these demonstrate that T lymphocytes subsets were affected by suppression. B lymphocyte gating was added to Figure S1. Doublet discrimination by scatter - which is the only relevant analysis in immunostaining by flow cytometry - is not possible in analogic equipments such as the one we used, a FACSCalibur. It is only capable of fluorescence doublet removal, which is only useful for DNA ploidy assessment.

Comments to editor:

I still believe the content of this paper is interesting and the data contribute to the field in a significant fashion. Rewording regarding the CD45+CD4+ and CD45+CD8+ cells in all appropriate locations is still needed. Thus, if the editor agrees, I believe the paper could be accepted with current experiments if appropriate revisions are applied.

June 27, 2020

Prof. Emanuel Maltempi de Souza
Federal University of Parana
Biochemistry and Molecular Biology
Curitiba, Parana 81531980
Brazil

Re: mSystems00080-20R2 (Cyclophosphamide increases Lactobacillus in the intestinal microbiota in chickens)

Dear Prof. Emanuel Maltempi de Souza:

Your manuscript has been accepted, and I am forwarding it to the ASM Journals Department for publication. For your reference, ASM Journals' address is given below. Before it can be scheduled for publication, your manuscript will be checked by the mSystems senior production editor, Ellie Ghatineh, to make sure that all elements meet the technical requirements for publication. She will contact you if anything needs to be revised before copyediting and production can begin. Otherwise, you will be notified when your proofs are ready to be viewed.

Sincerely,

Robert Beiko
Editor, mSystems

Supplemental Material: Accept

Figure S1: Accept

Supplemental Material: Accept

Supplemental Material: Accept

Figure S3: Accept

Figure S2: Accept